# A Study on the Oil-Bearing Stability of Salt-Resistant Foam and an Explanation of the Viscoelastic Phenomenon

**Changhua Yang** [1,2,*] **and Zhenye Yu** [2]

1 School of Petroleum Engineering, Xi'an Shiyou University, Xi'an 710065, China
2 Western Hypotonic-Special Hypotonic Reservoir Development and Treatment of Ministry of Education, Xi'an 710065, China; y_chenier@163.com
* Correspondence: ych569@126.com

**Abstract:** Foam is a medium-stable system composed of gas and liquid phases, which has the advantages of low density at the gas phase and high viscosity at the liquid phase, and has a wide application in oil and gas field development and mineral flotation, but its special medium-stable system also brings many problems in industry applications. Scientists have carried out extensive analyses and research on the foam stability and bubble-bursting mechanism, which initially clarified the rules of bubble breakage caused by environmental factors such as temperature and pressure, but the mechanism of bubble bursting under the action of internal factors such as liquid mineralization and oil concentration of the films is still not clearly defined. In this paper, we propose a compound salt-resistant foaming agent, investigated the influence of the aggregation and adsorption behavior of oil droplets on the liquid films and boundaries, and established a relevant aggregation and adsorption model with the population balance equation. We put forward a liquid film drainage mechanism based on the distribution, aggregation, and transport of oil droplets in the liquid films, so as to explain the changes in foam stability under the action of oil droplets. On the other hand, the viscoelastic analysis of foam fluid is performed with a rheometer, and the results show that in comparison with conventional power-law fluid, foam fluid has a complex rheological behavior for low shear thickening, but high shear thinning.

**Keywords:** polymer foam; foam stability; droplet aggregation; foam rheology

## 1. Introduction

Aqueous-based foams are composed of liquid-phase films and a gas-phase core, which have low density and high viscosity [1], enabling them to be widely used in many fields, such as petroleum and gas development [2], drilling and well workover operations [3], and froth flotation in the mining industry [4]. However, foams have a large specific surface and belong to thermodynamically unstable systems, which make it difficult for them to exist stably for a long time [5]. This problem creates numerous difficulties and challenges for its further application and extension in industrial applications. Currently, the factors affecting foams' stability are composed of two parts: the external environment in which the foams are located and the composition of the foaming agent liquid.

The earliest mechanism of bubble bursting was proposed with the effects of diffusion and evaporation, i.e., gas escaping from the gas nucleus and water evaporation from the liquid films. Further studies on evaporation and diffusion revealed that temperature and pressure are the key factors affecting foam stability [6], with higher temperature exacerbating the evaporation of water from the liquid film, and higher pressure effectively mitigating the escape of gas from the gas nucleus. Since then, it has also been pointed out that the presence of salt ions and hydrocarbons has a great influence on foam stability [7]. However, the related mechanism and the coarsening and aggregation process of foam under different external environments still need to be studied further [8].

Generally, foaming agent liquids are formed by compounding surfactants and polymeric stabilizers, which has a significant influence on stabilization. Foam stability relies on the number of branched chains and functional groups of the surfactants and polymers. Relevant studies have shown that it is necessary to obtain closely arranged molecular films to enhance the stability of films and foam [9]. As for polymeric stabilizers, on the one hand, the presence of polymers can relieve the film's drainage velocity, and on the other hand, it also enhance the strength of the foam through chemical linking such as hydrogen bonding. In addition, recent studies have shown that the addition of nanoparticles to foaming agent liquids can further improve stability by increasing the thickness of the liquid films [10,11].

In summary, this paper explores the mechanisms related to foam stability in the presence of brine ions and oil, and studies the dynamic phenomenon of foam liquid based on the rheological theory.

## 2. Materials and Methods

### 2.1. Materials

Materials were as follows:

Sodium dodecyl benzene sulfonate (SLS), chemically pure (Xingding Chemical Products Co. Zhengzhou, China);

Sodium $\alpha$-olefin sulfonate (AOS), chemically pure (Jurang Chemical Co. Guangzhou, China);

Cocoamidopropyl Betaine (CHSB-35), chemically pure (Xinwang Chemical Co. Dongying, China);

DP-1 stabilizer: the main component was anionic partially hydrolyzed polyacrylamide (HAPAM, a polymer) with a number average molecular weight of $1.6 \times 10^7$.

Simulated formation water: taking 96 g sodium chloride, 21 g calcium chloride, and 6.4 g hexahydrate magnesium chloride into 1000 mL deionized water, which obtained the simulated formation water with mineralization of $1.2 \times 10^5$ mg/L. The simulated formation water with different mineralization can be obtained by adding deionized water in different volumes;

Experimental oil: 0.9% kerosene + 0.1% 5# white oil;

HAAKE RS 300 rheometer (Thermo Fisher Scientific. Shanghai, China);

EM30-AX microscope (COXEM Inc. Beijing, China).

### 2.2. Methods

#### 2.2.1. Salt-Resistant Foam System

Three surfactants with different concentrations were poured into a Waring blender and stirred for 1 min at 1600 r/min. Then, the obtained foam was put into a measuring cylinder and the height of it was recorded at different times. The surfactant concentration that had the maximum foam value was used as the optimum foaming agent concentration. The different surfactants were mixed below the optimum concentration at a certain proportion and DP-1 stabilizer was added. The combination with the maximum foam volume can be studied with a Waring blender, referred to as the AC-5 foaming agent system. Simulated formation water with different mineralization was added to this system, and the foam was generated using the Waring blender. The volume of foam changing with time was observed and recorded, and the time taken to reduce the volume of foam to half is called the half-time. On this basis, whether the AC-5 foaming agent system is affected by the salt ions or not was evaluated.

#### 2.2.2. Oil Contained Stability

The experimental oil was mixed into AC-5 foaming agent with 0.3 wt.%, 0.5 wt.%, 0.7 wt.%, and 0.9 wt.%, The foam was generated by Waring blender and then loaded into a measuring cylinder. The evolution of foam volume and foam quality (volume of gas divided by the volume of foam) was recorded by the camera. Then, the foam was removed from the control valve into a transparent cuvette at different moments. To ensure the foam had a stable morphology during observation, the transparent cuvette was frozen in

liquid nitrogen. Then, the structure of foam and liquid films was observed with EM30-AX microscope. The images of the foam in the cuvette were acquired with computer, selecting the Plateau boundary where the liquid film was thicker for observation and increasing the objective magnification until the oil droplets were present in the monitor. The diameter of oil droplets and thickness of films were selected, and the volume ratio of oil droplets in the liquid films was calculated.

### 2.2.3. Viscoelastic Phenomenon

The viscosity of foaming agent and foam liquid are mainly controlled by the DP-1 stabilizer. The rheological parameters of foaming agent and foam liquid, with the different concentrations of DP-1 stabilizer, can be determined by the rheometer (HAAKE RS 300) in the linear viscoelastic area (0.1 Hz). The rheometer parameters (temperature, time, shear stress, etc.) were set to detect and evaluate the rheological behaviors of the foaming agent and foam liquid. The obtained data were plotted in the axes, and numerical fitting functions were used to determine the rheological equations of foam liquid according to the properties of curves.

The complete experimental procedure described in the Section 2.2 is shown in Figure 1.

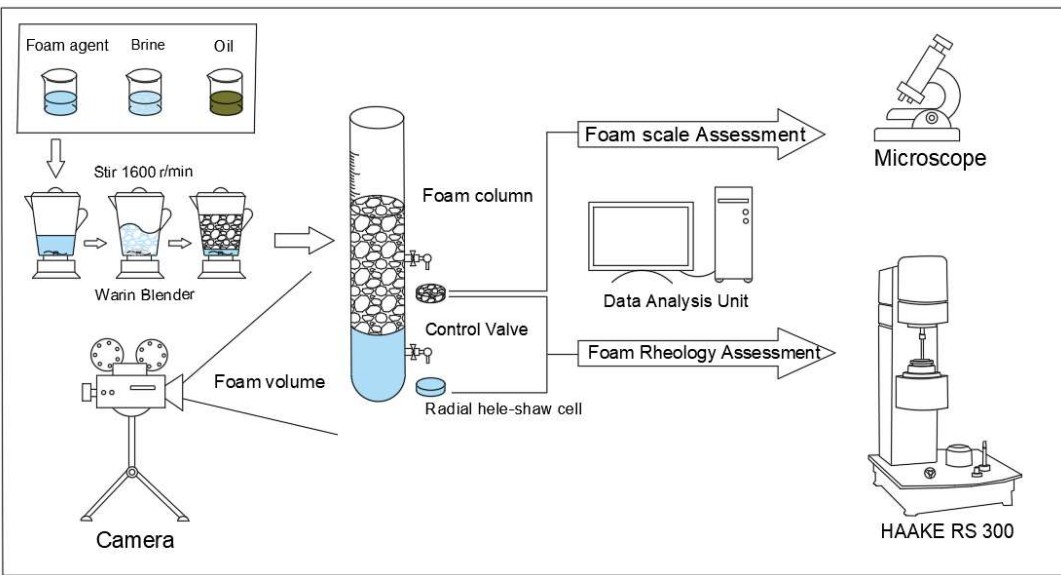

**Figure 1.** Experimental procedure of foam stability and viscoelasticity measurement.

## 3. Results and Discussion

### 3.1. Salt-Resistant Foam System

In order to determine the applicable concentration range of SLS, AOS, and CHSB-35, the foam volume and half-time were evaluated with a Waring blender with different concentrations, and the results are shown in Figure 2.

When the concentration is between 0.3 wt.% and 0.7 wt.%, the foam volume and half-time are positively related to the concentration; at this time, increasing the concentration can effectively improve the number of active molecules on the surface of liquid. While the concentration is more than 0.7 wt.%, the foam volume and the half-time do not increase with the concentration significantly. This is because when the concentrations of the surfactant reach critical micelle concentrations (CMC), and the number of active molecules on the surface and micelles in the solution keep a dynamic balance, the number of active molecules on the surface of the solution no longer grows significantly [12]. Further experiments have shown that the foaming agent system is obtained by a ratio of 1:5 in AOS and CHSB-35 and by adding the foam stabilizer DP-1 at 0.07 wt.%, which the system calls AC-5 and has the best foaming capacity, as shown in Figure 3.

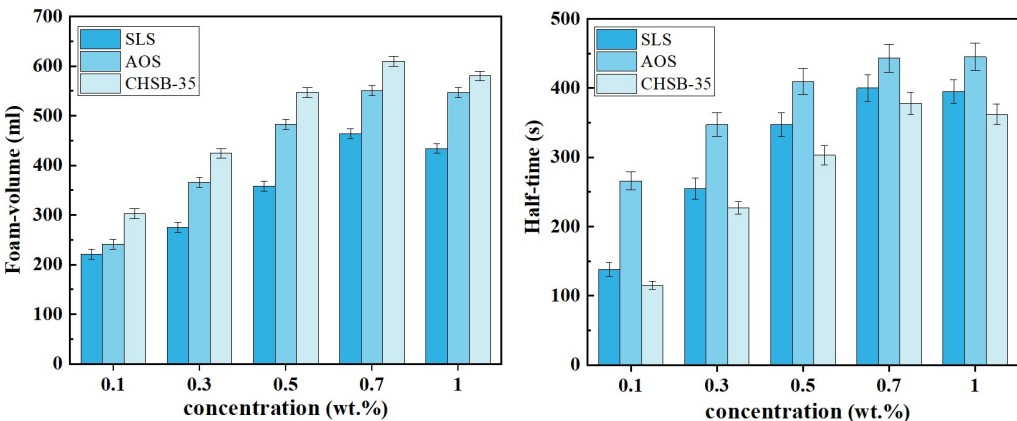

**Figure 2.** Relationship between surfactant concentration and foaming capacity.

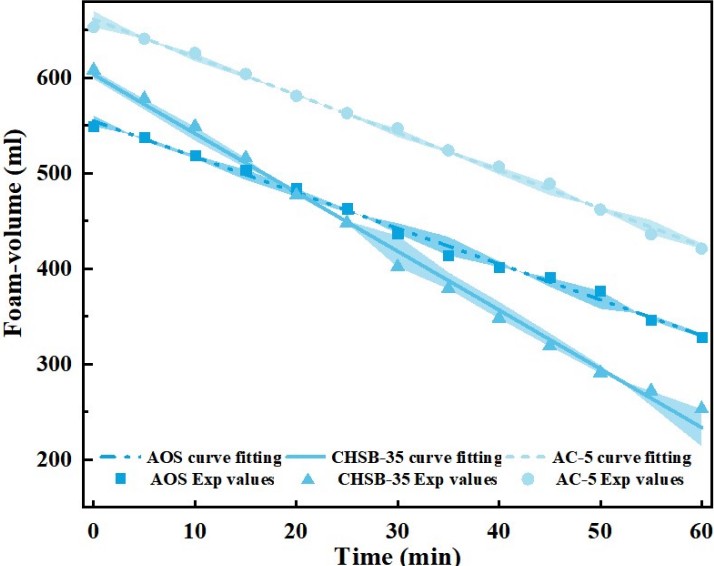

**Figure 3.** Evaluation of the foaming capacity of AC-5. The shaded part in the figure indicates the fitting error; the coefficient of determination $R^2$ is greater than 0.99.

According to the fitting results, the change in foam volume with time is a linear function. Compared with different types of surfactants alone, the foam produced by AC-5 also has a stable pattern of decay, which indicates that the components of the system are able to create dense films that cover the foam films, which have higher strength and the ability to restrain water evaporation in the films [13].

The AC-5 system was formulated by brine at different mineralizations, and the salinity tolerance was evaluated with a Waring blender. The AC-5 foaming capacity with different mineralizations of $1.2 \times 10^5$ mg/L, $0.8 \times 10^5$ mg/L and $0.4 \times 10^5$ mg/L is shown in Figure 4.

It can be seen from Figure 3 that the foam volume of the system can be influenced by salt ions [10], the lower concentrations of salt ions increase the volume of foam produced by foaming agents while higher concentrations have the opposite effect. The reason for this is that the main component of DP-1 is anionic HAPAM, which is negatively charged after dissolving in saline water at high levels as well as attracting the cations in the solution, creating an "ionic cross-linking" reaction in the polymer and leading to a "curled up" polymer in the solution. The ionic cross-linking reaction will cause the polymer in solution to "curl up" and reduce the viscosity of the liquid. On the one hand, it reduces the interfacial tension, which needs to be overcome when the foam generation makes the foam volume obviously increase [14]; on the other hand, it accelerates the bubble liquid films' drainage

rate, which reduces the foam stability. The number of molecules adsorbed in the solution surface increases and then decreases with ionic density [15]. The foam films' drainage velocity at different mineralizations was calculated based on the experiment, which is shown in Figure 5.

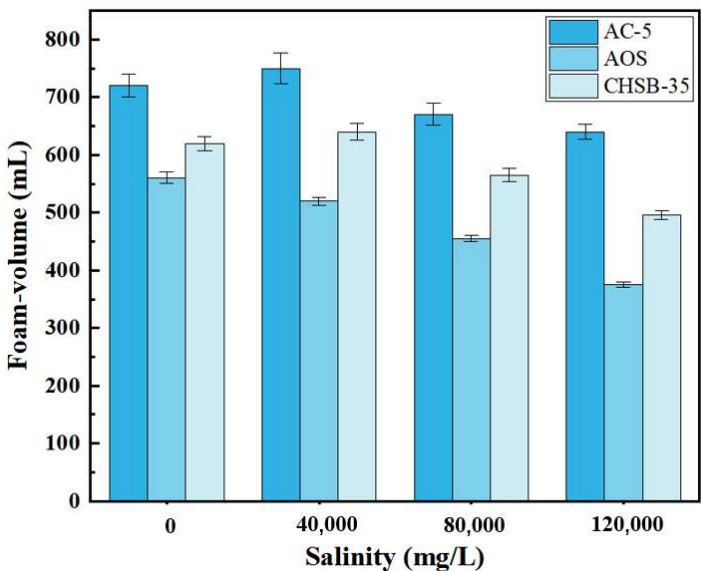

**Figure 4.** Evaluation of the salt-resistance of AC-5 compared with other foaming agents.

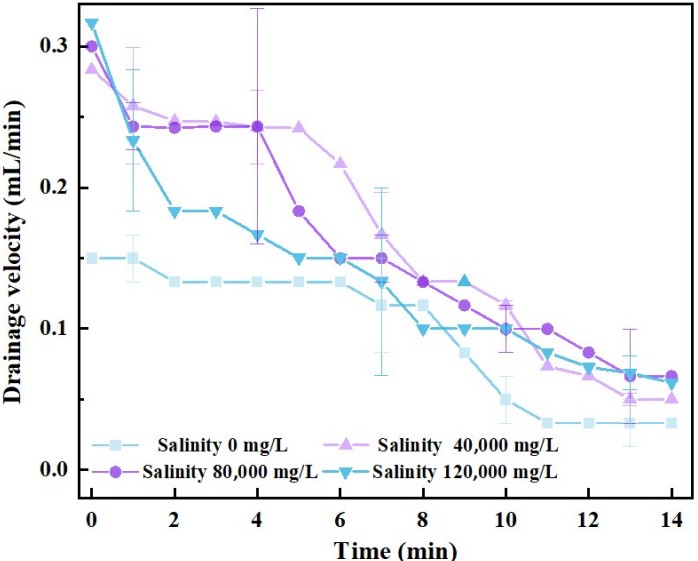

**Figure 5.** The AC-5 foam system films' drainage velocity at different mineralization.

The addition of brine ions increased the velocity of the films' drainage and diminished the effect of the DP-1 stabilizer. However, experimental results for different mineralizations indicate that the effect of brine ions on the foaming agent system was slight; this means that CHSB-35 belongs to an amphoteric surfactant, and its improved system has excellent salt resistance. In addition, the drainage velocity at different mineralizations was approximately similar before 1 min; this is because the drainage velocity at this stage was mainly related to the viscosity and radius curvature of the liquid films. When the liquid films thinned more and more, the diffusion double layer on the films stopped the thinning process by electrostatic repulsion, thus slowing down the drainage velocity and improving the foam stability [16]. This is the reason why there is a significant decrease in drainage velocity between 6 and 10 min in Figure 5.

### 3.2. Oil-Containing Stability Analysis

The AC-5 foaming agent system was mixed with experimental oil at different concentrations by high-speed stirring. The effect of experimental oil on the AC-5 foaming agent was evaluated with a Waring blender; the results are shown in Figure 6.

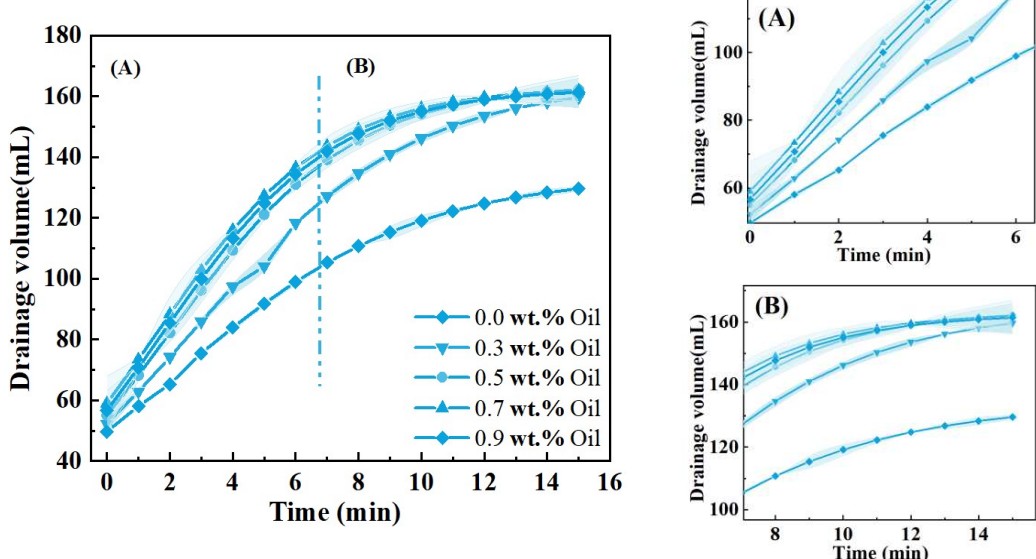

**Figure 6.** Effect of experimental oil on foam film drainage process in the AC-5 foaming system: (**A**) rapid drainage period; (**B**) approximate stabilization period. The shaded part in the figure indicates the fitting error. The coefficient of determination $R^2$ is greater than 0.97.

The slope of Figure 6 indicates the drainage velocity in the liquid films, which can be approximated as a linear function of two segments. The low concentration of experimental oil has a significant reduction in foam stability. This is because the high speed of stirring makes the oil and foaming agent exist as an emulsion form [17]. However, the experimental results at 0.5 wt.% and 0.9 wt.% do not differ significantly, which indicates that the AC-5 system has resistance to oil. In Figure 6A, the drainage velocity increases significantly after adding a small amount of oil, but the high concentration of oil shows a nearly consistent drainage velocity; in Figure 6B, oil-added foam has a lower drainage velocity than water-based foam. This means that the presence of oil has a specific "stabilizing" effect on the foam. This phenomenon can be explained based on the hypothesis of the distribution model for emulsified oil droplets in the foam films (Figure 7A,C) and Plateau borders (Figure 7B).

The oil droplets, which have a surfactant molecule surface layer, are mainly present in the films at two ways: suspended in the foaming agent films or Plateau borders; and attached to the boundary layer formed by the surfactant. The presence of oil droplets in Figure 6A accelerates the drainage velocity of the foam films, while in Figure 6B, the presence of oil droplets acts as a "stabilizer". This strange phenomenon may be produced by the movement of oil droplets on the films and Plateau borders.

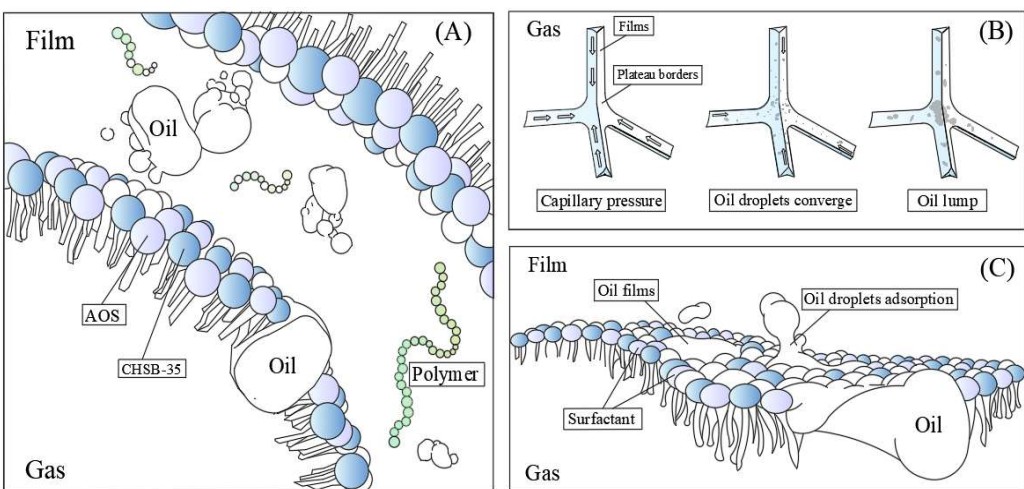

**Figure 7.** Hypothesis of distribution model for emulsified oil droplets. (**A**) Oil droplets distributed in the foaming agent and films; (**B**) coalescence formation of oil droplets in Plateau borders; (**C**) adsorption and coalescence of oil droplets at the boundary of liquid films.

The population balance equation, which is a continuous form of the numerical density function, is usually used to describe the motion in a series of small scales for discrete phases in a multiphase flow (e.g., microscopic pulsations in turbulent flows, aggregation of bubbles and emulsions, etc.). The establishment of the population balance equation can provide a method for solving the discrete term system on a macroscopic scale. Thus, the population balance equation (Equation (1)) can be a more precise mathematical method to describe the aggregation phenomena of emulsion oil droplets in the films.

$$\frac{\partial f(\zeta;x,\mathrm{t})}{\partial t} + \frac{\partial}{\partial x_i}\left( \langle u_i \rangle_\xi f(\zeta;x,\mathrm{t}) \right) = S(\zeta) \tag{1}$$

where $f(\zeta;x,\mathrm{t})$ describes the concentration distribution of discrete terms in time t, space x for different properties. <ui> indicates the average velocity of a particle at a certain attribute. $S(\zeta)$ is the source term, where $\zeta$ represents the properties of the oil droplet at a given location at a given time, including particle size, volume, characteristic length, etc. It is a row matrix, which represents the microscopic behavior (fragmentation, migration, and aggregation) of the discrete term particles. To simplify the model, we only consider the effect of the coalescence behavior of oil droplets on the numerical density function, which is written as Equation (2).

$$S(V) = \frac{1}{2}\int_0^V \beta(V-v,v)f(V-v;t)f(v;t)dv - f(V;t)\cdot\int_0^\infty \beta(V,v)f(v;t)dv \tag{2}$$

where $\beta(V,v)$ is the coalescence rate of particles with volume $V$ and $v$. The first term on the right-hand side of Equation (2) is the increment in the numerical density of the larger droplet (volume $V$) resulting from the aggregation of the smaller droplet (volume $v$), and the second term is the decrease in the numerical density of the larger droplet resulting from the aggregation of the droplet with other particles. In order to solve the equilibrium equation, the velocity function of particle motion and the coalescence rate function also need to be constructed.

The two oil droplets in the liquid film are close to each other by gravity, buoyancy, traction, capillary force, and additional resistance; according to the second Newton's law,

we can obtain the formula for the motion of two oil droplets close to each other, as shown in Equation (3).

$$
\begin{cases}
\frac{4\pi}{3} R_1^3 (\rho_d + C_{VM}\rho_c) \frac{dU_{d1}}{dt} = F_1 - \frac{\beta_1 \pi \sigma_f^2}{R_1} - F_{Dform,1} - \pi R_1^2 C_{D,1} \frac{\rho_c U_{d1}^2}{2} \\
\frac{4\pi}{3} R_2^3 (\rho_d + C_{VM}\rho_c) \frac{dU_{d2}}{dt} = F_2 - \frac{\beta_2 \pi \sigma_f^2}{R_1} - F_{Dform,2} - \pi R_2^2 C_{D,2} \frac{\rho_c U_{d2}^2}{2}
\end{cases}
\tag{3}
$$

where $R_1$ and $R_2$ are the radii of the two oil droplets; $\rho_c$ and $\rho_d$ are the densities of the continuous and discrete phases; $C_{vm}$ is the volume fraction of the discrete phases; $F_1$ represents the external force on the oil droplet; $\sigma_f$ is the surface tension; $F_{Dform}$ is the additional resistance; and $U_d$ is the relative rate between the two oil droplets. The relative velocity can be calculated by subtracting between the two equations (Equation (4)), which can be used as an approximate substitute for the particle average rate after neglecting the errors brought by the particle size.

$$
\begin{cases}
\frac{dU_{rel,mc}}{dt} = \frac{d\langle u_i \rangle}{dt} = \frac{3}{4} \frac{F_1 - \xi^3 F_2}{(\rho_d + C_{VM}\rho_C)\pi R_1^3} - F_\sigma - F_D \\
F_\sigma = \frac{3}{4} \frac{\sigma A R}{(\rho_d + C_{VM}\rho_C)\pi R_1^3} [(\beta_1 + 2) + (\beta_2 + 2)\xi^4] \\
F_D = \frac{3u_c}{16(\rho_d + C_{VM}\rho_c)R_1^2} (U_{d_1} C_{D,1} Re_1 - \xi^2 U_{d_2} C_{D,2} Re_2) \\
\xi = R_2 / R_1 \qquad (2-4) \\
\frac{dA}{dt} = \frac{d(r_f/R_1)^2}{dt} = \frac{2U_{rel,mc}}{R_1\left[(1-A)^{-1/2} + \xi(1-\xi^2 A)^{-1/2} - \frac{3}{4}(1+\xi^3)A\right]} \\
Re_1 = \rho_c d_1 U_{d_1}/\mu \qquad Re_2 = \rho_c d_2 U_{d2}/\mu_c \\
U_{d_1} = U_{mc} + \frac{U_{rel,mc}}{(1+\xi^3)} \qquad U_{d_2} = U_{mc} - \frac{\xi^3 U_{rel,mc}}{(1+\xi^3)} \\
\frac{dU_{mc}}{dt} = \frac{3\xi^3}{4(1+\xi^3)} \frac{F_1 + F_2}{(\rho_d + C_{VM}\rho_c)\pi R_1^3} - \frac{3}{16}\frac{\xi^2}{1+\xi^3} \frac{\mu_c}{(\rho_d + C_{VM}\rho_c)R_1^2} (\xi U_{d_1} C_{D,1} Re_1 + U_{d_2} C_{D,2} Re_2)
\end{cases}
\tag{4}
$$

where $\xi$ is the particle size ratio; $A$ is the dimensionless liquid film area; $Re$ is the Reynolds number of the fluid volume element; $U_{rel,mc}$ is the relative velocity of motion between two particle centers of mass after neglecting capillary forces and additional drag.

Taking the aggregation efficiency $\beta(V, v)$ proposed by Prince and Michael J (Equation (5)) [18] into (Equation (2)), the population balance equation for the consolidation of oil droplets in the foam films can be achieved.

$$
\begin{cases}
\beta(V, v) = \exp\left[-c_{p,1} \frac{\rho_c^{1/2}\varepsilon^{1/3}}{\sigma^{1/2}} \frac{(d_1 d_2)^{3/2}}{(d_1 + d_2)^{13/6}}\right] \\
c_{p,1} = \left(w\%_{oil} \times \int_0^t \frac{\sqrt{d_1^2 + d_2^2}}{h} dt\right)^{1/2}
\end{cases}
\tag{5}
$$

As for the nonlinear hyperbolic equation, it is often difficult to obtain the analytical solution, which can be solved by the time-driven Monte Carlo method, as suggested by Liffman [19]. To be convenient for analyzing, different positions of the liquid film were taken to solve the volume fraction of oil droplets changing with time, which are shown in Figure 8.

At the early stage of foam generation, the liquid film is thicker, and the finely crushed oil droplets are gradually integrated into larger droplets under the effect of surface tension and gravity. At this time, the oil droplets gradually move to the Plateau boundary under the capillary force (the oil concentration at points A, C, and D decreases with time, while the oil concentration at point B increases with time), and during this process, small oil droplets will also gradually collect into larger ones. This phenomenon accelerates the thinning of the liquid films; when a large number of oil droplets gather at the Plateau borders, they create an "oil lump", which blocks the channel and slows down the drainage of the liquid films (In Figure 8, the values of points A, B, C, and D all stabilize after 180 s). Furthermore, the experiment also indicated that the higher the oil content, the longer the process lasted (Figure 6B).

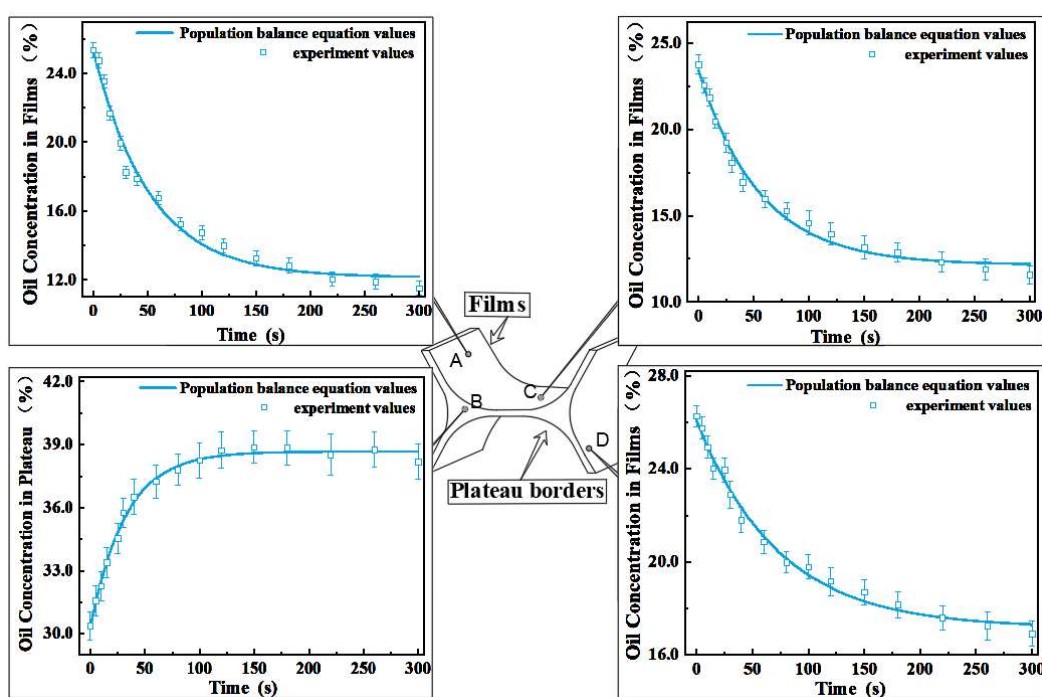

**Figure 8.** Comparison of the population balance equation and experimental: Points A, C, and D, the volume fraction of oil droplets in liquid films; point B, the volume fraction of oil droplets in Plateau borders.

After the establishment of the "oil block" at the Plateau borders, the oil begins to adsorb and accumulate on the surfactant boundary layer under the action of surface tension and volume force, which gradually destroys the original film structure and creates a new "oil film". This conclusion can be inferred from Figure 8; the concentration of oil at point B almost no longer changes after 180 s, while the oil concentration at points A, C, and D slowly decreases. When the area of "oil film" exceeds 15% of the surface area on a single bubble, the balance is broken, and the pressure of the internal gas forms a stress focus on the surface, thus causing the bubble burst [20]. It is worth noting that the impact by a single bubble bursting in this case will form a chain effect on the bubbles around it, and then cause a series of bubbles to burst at the same instant (instead of the bursting of the bubble columns at the gradual decline).

### 3.3. Interpretation of the Foam Fluid Viscoelasticity

The addition of the DP-1 stabilizer is not only able to increase the half-time [21], but also has a significant effect on the viscoelasticity and foam strength. The viscoelastic analysis of the AC-5 foaming agent (before stirring and not foaming) and foam fluid (after stirring) developed with different concentrations of DP-1 was carried out using a HAAKE RS 300 rheometer; the experimental results are shown in Figure 9.

The loss modulus is often used to characterize the fluid viscosity; the higher the loss modulus, the higher shear resistance of the fluid, and the higher the viscosity is. Comparing the two types of rheological curves, it is observed that when the concentration of DP-1 is up to 0.1 wt.%, the foam fluid can still retain a large viscosity under high shear stress. Compared with the foaming agent, the foam fluid has better viscosity and shear resistance. In addition, there are large distinctions between the rheological profiles of foam fluid in different DP-1 concentrations [13]. In the low concentration of the DP-1 foaming agent, the viscosity of foam fluid decreases with increasing shear stress, which is consistent with the shear dilution property of polymer solution. At a high concentration of foam fluid, there is a complex rheological relationship; with increasing shear stress, the loss modulus shows a rapid increase and then a slow decrease. In order to further investigate, the storage

modulus and loss modulus of the foam fluids at 0.1 wt.% DP-1 stabilizer were measured at low stress and fit according to the least-squares principle, respectively (Figure 10).

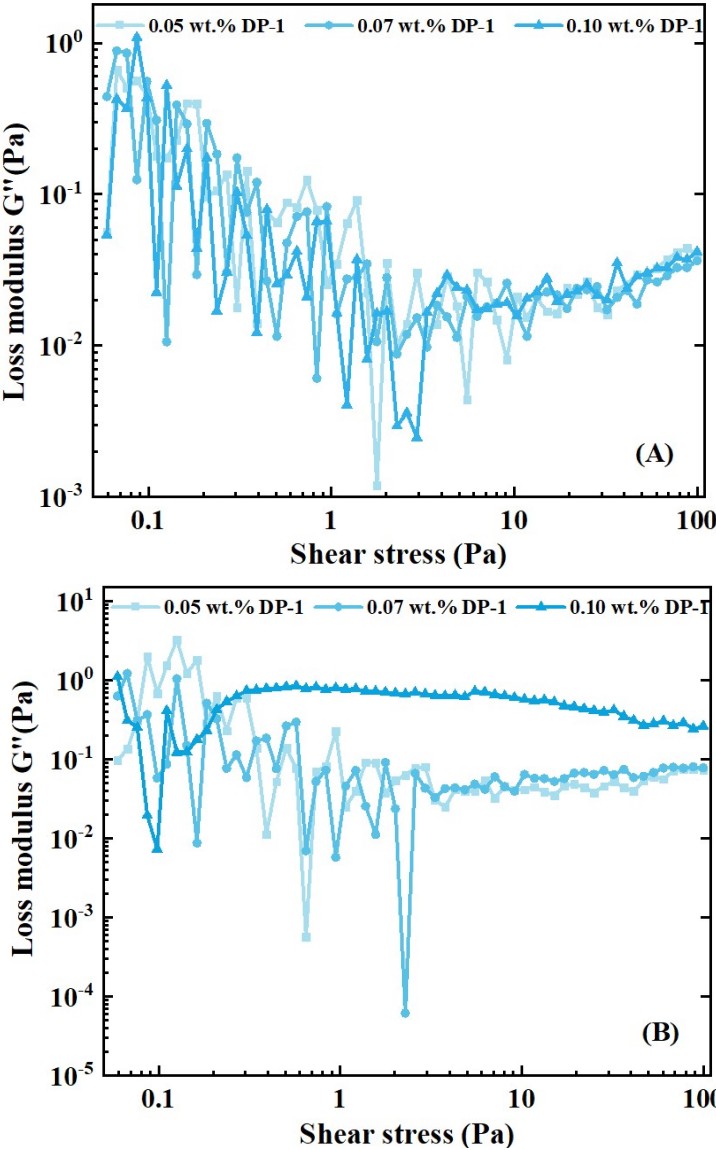

**Figure 9.** Effect of shear stress on loss modulus at different concentrations of DP-1 stabilizer. (**A**) The loss modulus of AC-5 changes with shear stress; (**B**) the loss modulus of foam fluid changes with shear stress.

At low frequency ($\leq 1$ Hz) and shear stress ($\leq 0.14$ Pa), the polymers in the DP-1 stabilizer are able to offset the stress force by changing their conformation. When the stress force is gone, the conformation of the polymers can be restored [22], i.e., when the storage modulus G′ is higher than the loss modulus G′, the elasticity of the foam plays a dominant role on a macroscopic scale. After being acted upon by an external force, the work of force will be stored as the elastic energy by the foam deformation. After removal of the external force, the foam will return to its original state [23]. Once the shear stress is obviously enhanced, the deformation is not enough to store the external work; thus, the foam fluid cannot maintain its stationary state, and excess energy will be stored in the form of internal frictional thermal energy. At that time, foam fluid starts flowing, and exhibits some viscosity on a macroscopic scale [24]. In order to have a more comprehensive understanding of foam rheology, we investigated the effects of shear times (Figure 11) and

shear rates (Figure 12) to liquid viscosity, and established the relative rheological models by exponential function, with the following results.

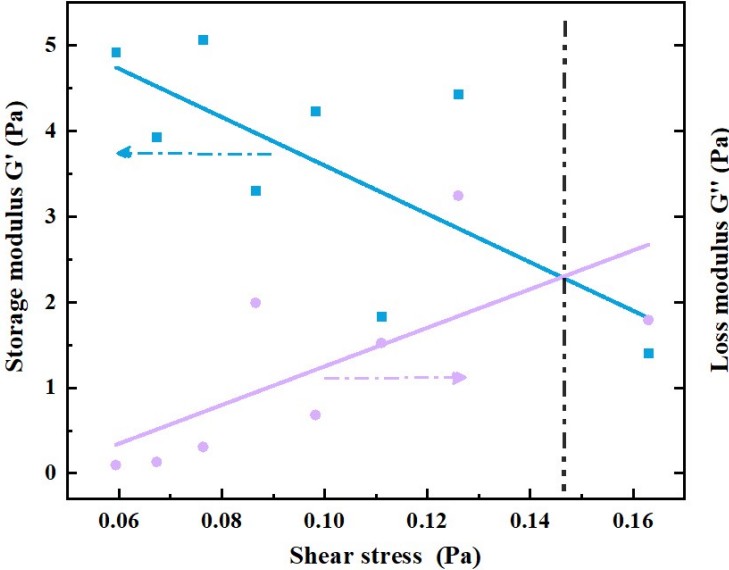

**Figure 10.** The storage modulus and loss modulus of foam fluid at low frequency and shear stress.

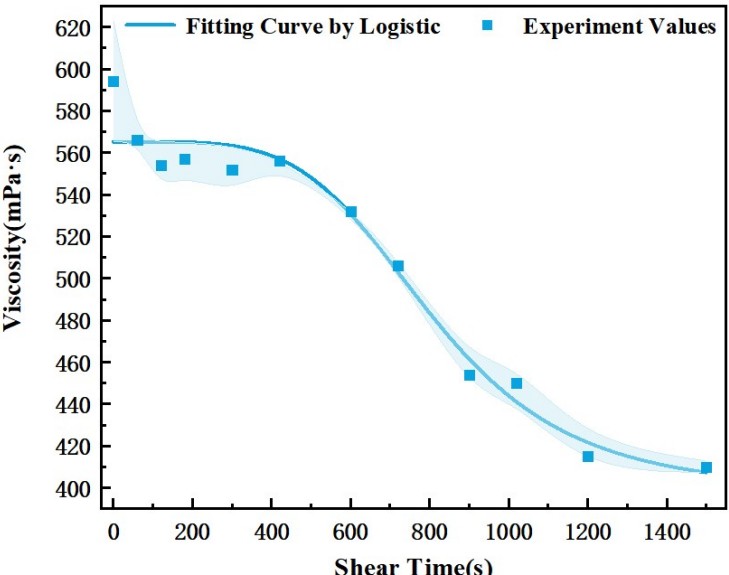

**Figure 11.** Effect of shear time on the viscosity of the foam fluid.

The logistic function was used to fit the experimental data of foam viscosity changes with the shear time; the results are shown in Equation (6).

$$\mu = 397.63 + 167.53 \, / \, \left( 1 + \left( \frac{t}{808.811} \right)^{4.52} \right) \tag{6}$$

Here, $\mu$ is foam fluid viscosity, in mPa-s; t is shear time, in seconds; the coefficient of determination $R^2$ is greater than 0.95, so the fitting results can explain experimental data well. When the shear rate is at a certain level, with the increase in shear time, the foam liquid exhibits shear dilution characteristics. On the one hand, the prolonged shearing will destroy the stability of films and accelerate the fragmentation of bubbles; on the other hand, it also distorts the conformation of the polymer in films. With the combination of these two factors, there is a gradual reduction in the viscosity of foam fluid.

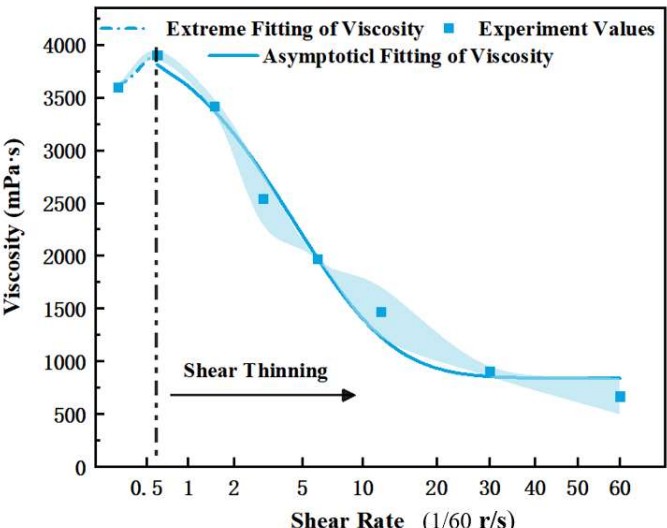

**Figure 12.** Effect of the shear rate on the viscosity of the foam fluid.

Experiments have shown that the relationship between foam fluid viscosity and shear rate is not a simple power law, as was believed for a long time. Instead, it shows the complex rheological behavior of low shear rate thickening and high shear rate dilution. To further describe this phenomenon, a function was adopted to fit the variation of foam viscosity with shear rate; the results are shown in Equation (7).

$$\begin{cases} \mu = 3599 + 312 \cdot \exp\left(e^{(-z)} - z + 1\right); z = (\gamma - 0.6) / 0.181; \gamma \le 0.7\,\mathrm{r/min} \\ \mu = 840.45 - (-3303.18) \times 0.8375^{\gamma}; \qquad \gamma > 0.7\,\mathrm{r/min} \end{cases} \quad (7)$$

where $\mu$ is the foam viscosity, in mPa-s; $\gamma$ is the shear rate, in r/s; the coefficient of determination $R^2$ is higher than 0.9. The fitting results have good ability to explain the experiment. When the shear rate is higher, the rheological law of the foam fluid is consistent with the power law, but when the shear rate is lower, the viscoelasticity behaves like the properties in low shear stress (Figure 9B).

### 4. Conclusions

1. The draining process of foam films at a high degree of mineralization is complicated, and the film draining velocity is reduced by a "step-type" law; this process involves many factors, such as the curvature of the liquid films, the thickness of the double-electric layer, chemical reactions, etc. It is difficult to use a single model for analysis and verification, and the main mechanism of ionics affecting foam stability and the liquid film draining process has not yet been clarified; it is necessary to analyze this in depth and establish a complete theoretical system.

2. The influence of experimental oil depends on its presence of the liquid films. When oil droplets exist in the films at tiny particle sizes, they are gradually gathered into oil lumps. Under the driving effect of surface and volume pressure, oil lumps will accumulate in the Plateau borders; such behavior also increases the draining process of the liquid films. With the accumulation of oil droplets on the Plateau borders, large-volume oil droplets will hinder the film draining process. The population balance equation is used to described this process efficiently. Minimal oil droplets may be adsorbed on the foam film surface and disrupt the stress distribution of films, which will lead to a "stepwise" breakup mode of the foam column. This phenomenon has not yet received sufficient attention from scholars.

3. Compared with the polymer solution, the foam fluid has a clearly elastic–plastic characteristic. At low values of shear and stress, foam liquids exhibit the characteristic of increasing viscosity in the shear; at high shear rates or stresses, the foam exhibits

the rheological characteristics of the power law. These findings can extend the understanding of power-law liquids in the academic community, and help to further investigate the pressure distribution and fluid mechanics of foam liquid from the perspectives of interface mechanics and rheology.

**Author Contributions:** Conceptualization, C.Y. and Z.Y.; methodology, Z.Y.; software, Z.Y.; validation, C.Y. and Z.Y.; formal analysis, Z.Y.; investigation, Z.Y.; resources, C.Y.; data curation, Z.Y.; writing—original draft preparation, Z.Y.; writing—review and editing, Z.Y.; visualization, Z.Y.; supervision, C.Y.; project administration, C.Y.; funding acquisition, C.Y. and Z.Y. All authors have read and agreed to the published version of the manuscript.

**Funding:** Jidong Oilfield: Key Research Project-Research on Leading Technology of Gas Drive in Shallow, Middle and Deep High Water Reservoirs and Application: JDYT-2019-JS-262; Xi'an Shiyou University: Graduate Student Innovation and Practical Ability Cultivation Programme: YCS22213040.

**Data Availability Statement:** All data openly available in a public repository.

**Conflicts of Interest:** The authors declare no conflict of interest.

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
