# Peer review of "A Study on the Oil-Bearing Stability of Salt-Resistant Foam and an Explanation of the Viscoelastic Phenomenon"

_processes, doi:10.3390/pr11092598_

Round 1

Reviewer 1 Report

26.07.2023.

Review of processes-2464567;

Study on oil-bearing stability of salt-resistant foam and explanation of viscoelastic phenomenon.

The following errors were found:

1. Point 2.1. materials: CHSB-35 surfactant not discussed (only SLS and AOS are described in the text).

2. Point 2.2.1 and 2.2.3, what is the DP-1 stabilizer (not described in the text).

3. Point 2.2.2 what does ,,simulated oil ‘’ mean?

4. Point 3.1. Is it about increasing the resistance of the foam to decomposition under the influence of salt?

5. On page 5, under Fig. 4, the mechanism of the reaction in question should be written.

6. Page 4-5 and fig 5: AC-5 appears. What is this? Please explain preferably in the Materils section.

7. Point 3.2 is AC5 should be AC-5.

8. In point 3.2. there is mention of a Waring stirrer, but the test methods did not include this stirrer (similar to HAAKE RS 300 rheometer, page 10). Describe this method.

9. Page 7 verse 15, there is no literature, from what position the fragment of the text was quoted.

10. Page 9 - citations are missing.

11. Page 10 verse 3, is the phrase "thought in common sense" needed here?

12. Conclusion 3: What polymer? This polymer is not discussed in the Materials section.

26.07.2023.

Review of processes-2464567;

Study on oil-bearing stability of salt-resistant foam and explanation of viscoelastic phenomenon.

The following errors were found:

1. Point 2.1. materials: CHSB-35 surfactant not discussed (only SLS and AOS are described in the text).

2. Point 2.2.1 and 2.2.3, what is the DP-1 stabilizer (not described in the text).

3. Point 2.2.2 what does ,,simulated oil ‘’ mean?

4. Point 3.1. Is it about increasing the resistance of the foam to decomposition under the influence of salt?

5. On page 5, under Fig. 4, the mechanism of the reaction in question should be written.

6. Page 4-5 and fig 5: AC-5 appears. What is this? Please explain preferably in the Materils section.

7. Point 3.2 is AC5 should be AC-5.

8. In point 3.2. there is mention of a Waring stirrer, but the test methods did not include this stirrer (similar to HAAKE RS 300 rheometer, page 10). Describe this method.

9. Page 7 verse 15, there is no literature, from what position the fragment of the text was quoted.

10. Page 9 - citations are missing.

11. Page 10 verse 3, is the phrase "thought in common sense" needed here?

12. Conclusion 3: What polymer? This polymer is not discussed in the Materials section.

Author Response

Dear Reviewers,

Thanks very much for taking your time to review this manuscript. I really appreciate all your comments and suggestions! Please find my itemized responses in below and my revisions/corrections in the re-submitted files.

Appended to this letter is our point-by-point response to the comments raised by the reviewers. The comments are reproduced and our responses are given directly afterward in a different color (RED).

  • Point 1: Point 2.1. materials: CHSB-35 surfactant not discussed (only SLS and AOS are described in the text).

Response: I am sorry that the omission of the author has affected the reading, the Chemical name of CHSB-35 ( Cocoamidopropyl Betaine ) and related information has been added to the article.

  • Point 2: Point 2.2.1 and 2.2.3, what is the DP-1 stabilizer (not described in the text).

Response: DP-1 stabilizer, The main component is anionic partially hydrolysed polyacrylamide (HAPAM) with the Number-average Molecular Weight of 1.6 x 107. The main components of DP-1stabilizer have been added to the text. Thanks to the reviewer's work for making the article complete!

  • Point 3: Point 2.2.2 what does ‘’ simulatedoil ‘’ mean?

Response:  The ‘’ simulated oil ‘’ means that the special oil used in the experiment, which has more Transparency than black oil. Employing this oil will facilitate further observation by microscopy. Now, the authors recognizes that this statement may not be easy to understand, we have replaced the ‘’ simulated oil ‘’  with  ‘’ experimental oil ‘’ in the paper .We hope this change will lead to a better reading experience.

  • Point 4: Point 3.1. Is it about increasing the resistance of the foam to decomposition under the influence of salt?

Response: Point 3.1. This section starts with the compounding of different types of surfactants to obtain foaming agents with better performance. Then,focuses on how the presence of salt ions affects the stability of foam. It demonstrates that the presence of salt ions is the main factor influencing the bubble stability only at a certain stage(Fig5), and analyses the reasons for this phenomenon through the change in volume of the foam at different mineralisation levels.

  • Point 5: On page 5, under Fig. 4, the mechanism of the reaction in question should be written.

Response: Under Figure 4, the mechanism has been described in several ways, including the effect of salt ions on the surfactant at different concentrations and on the HAPAM in DP-1, which the authors believe is complete and has been extensively discussed in related papers. What’s more, In view of the priority and length of the manuscript, the authors wished to focus on the explanation of the stability and rheological phenomena of oil-bearing foams in the latter part of the paper.

  • Point 6: Page 4-5 and fig 5: AC-5 appears. What is this? Please explain preferably in the Materils section.

Response: By mixing the different types of surfactants mentioned in the “material” at a certain concentration and adding the stabilizer DP-1, a new foam agent will be obtained as AC-5. As this section does not involve new surfactants or other chemicals, the authors have not mentioned it in the material section. Also, after full consideration of the reviewers' comments, the naming of this Foaming agent system was added when the main components of AC-5 were first mentioned in the paper ( Page 4, first paragraph, last line ).

  • Point 7: Point 3.2 is AC5 should be AC-5.

Response: Thank you for your kind reading, the author has carefully checked the symbolic markings in the paper and corrected any incorrect ones.

  • Point 8: In point 3.2. there is mention of a Waring stirrer, but the test methods did not include this stirrer (similar to HAAKE RS 300 rheometer, page 10). Describe this method.

Response: Thanks for your suggestion! According to your opinion, the relevant experimental steps have been added to the manuscript in sections "2.2.1" and "2.2.3".

  • Point 9: Page 7 verse 15, there is no literature, from what position the fragment of the text was quoted.

Response: This part of the text is presented by the authors. With observation the movement of oil droplets in the foam film and the change of drainage rate of the film. On this basis, preliminary modelling and numerical calculations have been carried out to explain the mechanisms involved in the bursting of oil-containing foams. The inverted commas in the text were incorrectly edited by the authors and have been removed.

  • Point 10: Page 9 - citations are missing.

Response: Thank you for your comments, the missing citations have been re-edited.

  • Point 11: Page 10 verse 3, is the phrase "thought in common sense" needed here?

Response: At the authors' consideration, the word "thought in common sense" does seem to be inappropriate here,and already revised in the manuscript, so thank you for your valuable comments!

  • Point 12: Conclusion 3: What polymer? This polymer is not discussed in the Materials section.

Response: The polymer is anionic partially hydrolysed polyacrylamide (HAPAM) has been added to the material section as per the reviewer's comments.

We would like to thank the referee again for taking the time to review our manuscript!

Reviewer 2 Report

The manuscript “Study on oil-bearing stability of salt-resistant foam and explanation of viscoelastic phenomenon” by Yang Changhua and Yu Zhenye presents results of the experimental and theoretical studies of mechanisms controlling the stability of oil-containing foams. The research topic is interesting and consistent with the focus of the “Processes” journal. However, the research methodology, the interpretation of the data obtained and the style of their presentation encounter a number of questions and comments that do not allow me to recommend the manuscript for publication in its present form. They are listed below.

1. Section 2 “Materials and Methods” needs to be substantially revised and expanded. As the example, the statements “The optimum foaming concentrations ….” and “The simulated oil is poured into the foam agent fluid ..” do not provide any valuable information regarding the experimental conditions.

Another example: “the evolution of foam height and foam quality is recorded by the camera, the liquid and the bubbles are taken out at different moments”. What is a “foam quality” and how it was quantified? What does it mean -  “the liquid and the bubbles are taken out at different moments”? It is necessary to note that microscopic inspection of the foams in transparent cuvettes should take into account the difference between the bubble size distributions and liquid content in the shallow layer of the foam contacting with the cuvette wall and imaged by a microscope and in the underlying bulk of the foam. Accordingly, the details of foam structure inspection should be thoroughly explained. The examples of raw images illustrating the discussed behavior of oil droplets in liquid films are greatly appreciated. The special attention should be paid to characterization of the “changes of foam diameter and film thickness”; this point requires a special accuracy because of probable pitfalls. By the way, that is the “foam diameter”; maybe, it is the “bubble diameter”?

2. The shape and dimensions of Plateau-Gibbs channels and channel grid nodes are strongly influenced by the liquid content of the foam. Accordingly, the characteristics of the channel system change dramatically from wet to dry foam. The manuscript does not contain any information about the volume fractions of the liquid phase in the studied foams, which is important from the point of view of analyzing the structure of the network of Plateau-Gibbs channels. Additional references regarding this item are highly appreciated.

3. The authors often use very specific terminology without any comments or explanations. For example, Figure 2 and the caption to it. It is not clear what the “foam capacity” (the y-axis in the left panel is marked as "foaming ability").  What process is characterized by the "half-time" parameter? Relevant comments should be given in all similar cases throughout the text.

4. Figs 3 and 6. The confidence levels used to create the “shaded parts” in the graphs should be specified.

5. The sentence “As a result of the fittings, the foam volume was in good agreement with the first

order hydrodynamic equation in the condition of bubble burst” (below Fig. 3) is very mysterious. What “the first order hydrodynamic equation” is and where does it come from? What the “condition of bubble burst” is? Please, clarify this sentence in much more details. The additional formulas and relevant references with appropriate comments are very welcome.

6. The sentence below Fig. 4 with the term “the bubble height of the system”. It seems that the authors see no difference between foam bubbles and foam volume.

7. The theoretical model meets a lot of questions. First, it is necessary to introduce an explanatory figure and describe the physical origin for Eq. 2.1. Is the x-coordinate measured along an arbitrarily chosen Gibbs-Plateau channel or in some other way? Why the source term does not depend on time and coordinate but only on an arbitrarily introduced parameter? In Eq. 2-2, is the function f() the same as in Eq. 2-1? If so, there is a mismatch between the parameters of this function in Eqs (2-1) and (2-2). A lot of terms in Eqs. 2-1 – 2-5  is not specified. The system (2-5) is unclear. Does the first equation define the constant value or a time-dependent function? Similar examples can be continued, therefore it is proposed to radically rework this part of the manuscript for better understanding by the journal readers.

8. The simulation results (Fig. 8). The results are given in percentages, and the question immediately arises - with respect to what? In my opinion, these results are unclear and should be transformed to more physically sensing units (e.g., the volume concentration of droplets in films and surface concentration onto the surfaces). Besides, how the experimental data were obtained? (see the first comment).

9. The English writing is sometimes hard to follow. Substantial literary editing is necessary.

Extensive editing of English language required.

Author Response

Dear Editor and Reviewers,

Thanks very much for taking your time to review this manuscript. I really appreciate all your comments and suggestions! Please find my itemized responses in below and my revisions/corrections in the re-submitted files.

Appended to this letter is our point-by-point response to the comments raised by the reviewers and editors. The comments are reproduced and our responses are given directly afterward in a different color (RED).

  • Point 1: Section 2 “Materials and Methods” needs to be substantially revised and expanded. As the example, the statements “The optimum foaming concentrations ….” and “The experimental oilis poured into the foam agent fluid ..” do not provide any valuable information regarding the experimental conditions.
  • Another example: “the evolution of foam height and foam quality is recorded by the camera, the liquid and the bubbles are taken out at different moments”. What is a “foam quality” and how it was quantified? What does it mean -  “the liquid and the bubbles are taken out at different moments”? It is necessary to note that microscopic inspection of the foams in transparent cuvettes should take into account the difference between the bubble size distributions and liquid content in the shallow layer of the foam contacting with the cuvette wall and imaged by a microscope and in the underlying bulk of the foam. Accordingly, the details of foam structure inspection should be thoroughly explained.The examples of raw images illustrating the discussed behavior of oil droplets in liquid films are greatly appreciated. The special attention should be paid to characterization of the “changes of foam diameter and film thickness”; this point requires a special accuracy because of probable pitfalls. By the way, that is the “foam diameter”; maybe, it is the “bubble diameter”?

Response: Thank you for your valuable comments. After careful consideration, the authors have re-edited the original “methods ”section of the manuscript and have focused on the reviewers' suggestions in the editing process.

The term "foam quality" refers to the volume of gas in the foam divided by the volume of the foam and is used in the manuscript as an important parameter to measure the dryness of the foam."The expression " the liquid and the bubbles are taken out at different moments" may be problematic to understand and the authors have revised this part as “Then the foam was removed from the control valve into a transparent cuvette at different moments. ”

In the re-editing manuscript, the experimental steps for image acquisition by microscopy were described carefully, and considering the varying morphology of the foam in the transparent cuvette, a stepwise selection method was used to choose samples that were easy to observe. As for “changes of foam diameter and film thickness”, the authors concluded that, according to previous conclusions, the foam has a relatively long stabilization period (more than 40 min), and during the observation of oil droplet motion in the liquid films (0-10 min), the thickness of the films and the diameter of the foam do not change significantly, so the effect of this part on the experiment can be ignored. Variations in the film thickness are evaluated in the manuscript by looking at both the drainage rate and drainage volume in macro perspective.

  • Point 2: The shape and dimensions of Plateau-Gibbs channels and channel grid nodes are strongly influenced by the liquid content of the foam. Accordingly, the characteristics of the channel system change dramatically from wet to dry foam. The manuscript does not contain any information about the volume fractions of the liquid phase in the studied foams, which is important from the point of view of analyzing the structure of the network of Plateau-Gibbs channels. Additional references regarding this item are highly appreciated.

Response: The manuscript uses the parameter "foam mass" to describe the dryness of the foam, the higher the foam mass, the higher the gas content of the foam and the more the foam is inclined to be " Dry Foam ". The foams studied in the manuscript are all in the category of Dry Foams (foam quality above 0.7, and will not change significantly during the research process). According to Plateau's theorem (Joseph Plateau, 1801-1883), the larger the number of bubbles, the more complex the boundary problem, which involves a series of complex mathematical problems such as Differential geometry and Geometric measure theory. For the solution of complex Plateau boundary problems (especially in the case of a disordered accumulation of multiple irregular bubbles), the authors also hope to communicate and explore further in future research. The following diagram shows the structure of the Plato boundary for the stacking of 1-10 bubbles found by the authors in the course of their search for information. The authors believe that it would be interesting to further investigate the regularity of this based on the principle of minimum free energy (minimum surface area of the foam cluster)

Unfortunately, limited by the above-mentioned problems, the Plateau boundary can only be treated in the simplest geometric structure in the model of the manuscript(Fig 7(B) and Fig 8), and the authors consider this to be one of the main directions for further refinement of the work in the future.

  • Point 3: The authors often use very specific terminology without any comments or explanations. For example, Figure 2 and the caption to it. It is not clear what the “foam capacity” (the y-axis in the left panel is marked as "foaming ability").  What process is characterized by the "half-time" parameter? Relevant comments should be given in all similar cases throughout the text.

Response: Thank you for your suggestion. In response to this question, the authors have replaced "foaming capacity" with "foam volume" for ease of reading, as both in fact describe the same issue. The descriptions of "Half-time" and "foam quality" will be given in the section "Materials and methods".

  • Point 4: Figs 3 and 6. The confidence levels used to create the “shaded parts” in the graphs should be specified.

Response: According to the comments of the reviewers, the confidence levels was added at the corresponding position in the re-edited manuscript.

  • Point 5: The sentence “As a result of the fittings, the foam volume was in good agreement with the firstorder hydrodynamic equation in the condition of bubble burst” (below Fig. 3) is very mysterious. What “the first order hydrodynamic equation” is and where does it come from? What the “condition of bubble burst” is? Please, clarify this sentence in much more details. The additional formulas and relevant references with appropriate comments are very welcome.

Response: Thank you for your careful reading! This expression in the manuscript "the first order hydrodynamic equation" might be problematic and should be changed to "a linear function". For ease of understanding, the sentence should read "According to the fitting results, the pattern of change in foam volume with time is a linear function."

  • Point 6: The sentence below Fig. 4 with the term “the bubble height of the system”. It seems that the authors see no difference between foam bubbles and foam volume.

Response: While editing the manuscript, the author did confuse the concepts of words "foam bubbles" and "foam volume", to ensure that this does not create a barrier to reading, the words "foam bubbles " has been replaced by "foam volume". Thank you for raising this issue!

  • Point 7: The theoretical model meets a lot of questions. First, it is necessary to introduce an explanatory figure and describe the physical origin for Eq. 2.1.
  • Is the x-coordinate measured along an arbitrarily chosen Gibbs-Plateau channel or in some other way?
  • Why the source term does not depend on time and coordinate but only on an arbitrarily introduced parameter?
  • In Eq. 2-2, is the function f() the same as in Eq. 2-1? If so, there is a mismatch between the parameters of this function in Eqs (2-1) and (2-2). A lot of terms in Eqs. 2-1 – 2-5  is not specified.
  • The system (2-5) is unclear. Does the first equation define the constant value or a time-dependent function? Similar examples can be continued, therefore it is proposed to radically rework this part of the manuscript for better understanding by the journal readers.

Response: 

After considering the comments of reviewer, the authors believe it is important to add the physical origin and relevant background information before using the population balance equation. This part has already been reflected in the re-edited manuscript.

During the simulation, the x-coordinate is chosen to be the same as the Gibbs-Plateau channel in the horizontal direction, while the z-coordinate is the same as the direction of gravity.

where  ζ  represents the properties of the oil droplet at a given location at a given time, including particle size, volume, characteristic length, etc. It is a row matrix. This part has been modified in the manuscript to eliminate problems in the reading process.

In Eq. 2-2, the function f() is the same as in Eq. 2-1, The position coordinate x is missing from equation 2-2 which is because the position of the source term is given before the simulation and does not appear as an unknown parameter in equation 2-2. In response to the reviewer's point that there are numerous undefined parameters in Eqs. 2-1 to 2-5, the authors have added them to the manuscript.

The first equation is an expression for the aggregation efficiency of oil droplets, while the second equation gives the calculation of the parameter cp,1 in the first equation. To facilitate the reader's understanding, the parameter β(V,v) is used to express the aggregation efficiency in the revised manuscript.

  • Point 8: The simulation results (Fig. 8). The results are given in percentages, and the question immediately arises - with respect to what? In my opinion, these results are unclear and should be transformed to more physically sensing units (e.g., the volume concentration of droplets in films and surface concentration onto the surfaces). Besides, how the experimental data were obtained? (see the first comment).

Response: The percentages in the results indicate the volume fraction of oil droplets at different locations in the liquid films and Plateau boundary. The volume fraction of oil droplets at the observed locations,which is taken as the ratio between the volume of oil droplets and the volume of liquid in the corresponding liquid film. Finally, the experimental results are compared with the simulation results. The misstatement in this section has been corrected in the revised manuscript. The acquisition of the data is supplemented in the section "Methods".

  • Point 9: The English writing is sometimes hard to follow. Substantial literary editing is necessary.

Response: We sincerely apologize for the inconvenience of reading caused by grammatical problems and typos. Accordingly, we have corrected the problems and all similar ones throughout the manuscript without altering the paper’s original meaning.

We would like to thank the referee again for taking the time to review our manuscript!

Reviewer 3 Report

Often the spaces after the comma are lost or doubled, sometimes capital letters are used after the comma and so on, to support the authors, I have highlighted this part in yellow, please check through the text and figures!

Furthermore, instead of using several terms or expressions that are different from each other, it is necessary to establish one and keep it in the exposition of the text, in example: Figure b, Fig. B or Figure (b), you should make a choice and maintain it in the text.

Introductions

What are the “brine icons”? I searched in the web but couldn’t find any information about it.

“the perspective of rheology” sound scientifically strange, please change!

materials and methods

The section 2.1 Materials is unclear and should be redrafted to make it complete.

The dealer should be expressed within parentheses: (Henan Xingding Chemical Products Co.).

HAAKE RS 300 rheometer, Thermo Fisher Scientific. EM30-AX microscope, COXEM Inc. are the instruments used for the characterization? Please describe better.

The steps taken in preparing the solutions are confusing!

The section 2.2. DP-1 stabilizer appears but is not present in section 2.1.

What is it? A commercial product? More details are mandatory for reproducibility of experiment by third parties!

What is “simulated ground water”? Where is the preparation procedure described? Why do you use it instead of the deionized water? Please give to the reader more details or references.

If “brine ions” corresponds to the “ground water” or “simulated formation water”, please chose one of them and keep it in the text to avoid confusion in the reader.

2.2.2. what means “Oil Contained Stability”? What is “simulated oil”? What about the formulation of Simulated oil? Probably it is 0.9% kerosene and 0.1 % of 5# (!?!) white oil and called as “simulated condensate oil”.

Overall, the “materials and methods” section needs more clarity, the formulations and components used are not clear, for example which hydrocarbon is used as oil?

Results and discussion

Last four lines at Pag. 4. Author said: “Compared with different types of surfactants alone,” but where is reported the behavior of each foaming agents "alone"? Can they be added to figure 3?

it would also be interesting to show the behavior of the solutions without the DP1 stabilizer, as a term of comparison!

Where is AC-5 system defined?

Could you explain the phrase “It can be seen from the Figure 3 that the bubble height of the system can be increased by adding low concentrations of salt ions in a certain extent [10], while the bubble height of the system starts to decrease after further increasing the salt ion concentration.”,  I can’t understand if figure 3 and 4 are compared to show the foaming ability or somethings else.  “Bubble height” is the foaming ability or has another definition?

“anionic HPAM” should be defined in materials and method if use or the first time you use it.

Figure 2 and more general in the text. Weigh percent is expressed as “wt.%” not “% wt” or other.

Figure 6. The labels (on axes and inside the graph) are too small.

Figure 8. The labels (on axes and inside the graph) are too small.

Figure 9 a and b. The labels (on axes and inside the graph) are too small, keeping into account that they will be reduced in size for publishing, it is better make them larger.

Also, the colors should be changed, as well as the size and symbols to improve readability.

Figure12. Because on the x-axis the unit is seconds, the y-axis unit should also be in seconds instead of minutes! Check “(r/min)” and “Asymptoticl” in the label inside the graph!

The draft seems interesting, in broad terms it is clear what the authors intend to describe, but unfortunately some sections are confused in the construction of the text and should be reviewed to allow the reader to understand the development of the experimental parts and the processing of the data adopted for the discussion, the "conclusion" section should also be revised.

According to the definition of polymer as a macromolecule obtained by repeating a monomer, therefore oils and kerosene are not, which are the polymers considered/used by the authors in the work presented?

The English language is poor, it needs to be improved both in the form in the grammatical form and in the spelling.

Author Response

Dear Reviewers,

Thanks very much for taking your time to review this manuscript. I really appreciate all your comments and suggestions! Please find my itemized responses in below and my revisions/corrections in the re-submitted files.

Appended to this letter is our point-by-point response to the comments raised by the reviewers. The comments are reproduced and our responses are given directly afterward in a different color (RED).

  • Point 1: Often the spaces after the comma are lost or doubled, sometimes capital letters are used after the comma and so on, to support the authors, I have highlighted this part in yellow, please check through the text and figures!

Response: Thank you for your valuable comments, unfortunately the authors did not find the helpful manuscript that you provided on the website, but based on your suggestions, the authors have tried to check and correct the punctuation throughout the text, and if during the subsequent review process, any omissions are found, we hope you can contact the authors to further improve the manuscript! As a side note, some of the commas appear to be of different lengths on the spaces, which has been affected by the alignment of the text to give the manuscript a more pleasing typography.

  • Point 2: Furthermore, instead of using several terms or expressions that are different from each other, it is necessary to establish one and keep it in the exposition of the text, in example: Figure b, Fig. B or Figure (b), you should make a choice and maintain it in the text.

Response: This is truly a good suggestion for reader-friendly reading! Therefore the authors decided to use the logos (A) and (B) as terms and expressions for the manuscript.

Introductions

  • Point 3: What are the “brine icons”? I searched in the web but couldn’t find any information about it.

Response: I am sorry for the author's error in editing, the word "brine icons" should be written as "brine ion", and the authors have corrected this error in the re-edited manuscript.

  • Point 4: “the perspective of rheology” sound scientifically strange, please change!

Response: Thank you for your comments, the manuscript has been revised to "based on the rheological theory"!

materials and methods

  • Point 5: The section 2.1 Materials is unclear and should be redrafted to make it complete.

Response: To ensure that the “materials” section has a clear structure, this part has been re-edited to show the chemicals, the experimental oil composition and the simulated formation water formulation in separate paragraphs.

  • Point 6: The dealer should be expressed within parentheses: (Henan Xingding Chemical Products Co.).

Response: Thanks for your comments, this part of the manuscript has been re-edited.

  • Point 7: HAAKE RS 300 rheometer, Thermo Fisher Scientific. EM30-AX microscope, COXEM Inc. are the instruments used for the characterization? Please describe better.

Response: To give the reader a better understanding of the experiments in the manuscript and the work which the instrument involved in, this information is represented in the re-edited “methods”.

  • Point 8: The steps taken in preparing the solutions are confusing!The section 2.2. DP-1 stabilizer appears but is not present in section 2.1.What is it? A commercial product? More details are mandatory for reproducibility of experiment by third parties!

Response: DP-1 stabilizer, The main component is anionic partially hydrolysed polyacrylamide (HAPAM) with the Number-average Molecular Weight of 1.6 x 107. The main components of DP-1stabilizer have been added to the Manuscript. Thanks to the reviewer's work for making the article complete!

  • Point 9: What is “simulated ground water”? Where is the preparation procedure described? Why do you use it instead of the deionized water? Please give to the reader more details or references.
  • If “brine ions” corresponds to the “ground water” or “simulated formation water”, please chose one of them and keep it in the text to avoid confusion in the reader.

Response: The "simulated ground water" refers to brine formulated at a certain level of mineralisation according to the ionic concentration in the formation water and was used to evaluate the stability that the foam would have when exposed to different salt ions ( this part of the experiment takes into account the possibility of the foam contacting the ground water at the bottom of oil well). The preparation process is described in the section "Materials" of the revised manuscript.

The term "simulated formation water" refers to brine prepared in the laboratory according to the ionic composition of "ground water". In order to avoid unnecessary confusion for the reading public, the revised manuscript decided to use the term "simulated formation water" instead of "ground water". This was done because "simulated formation water" and "ground water" are identical only in terms of mineralisation(In another word, the "ground water" may have other substances e.g. micro-organisms, bacteria, acids, etc.)

  • Point 10: 2.2. what means “Oil Contained Stability”? What is “experimental oil”? What about the formulation of experimental oil? Probably it is 0.9% kerosene and 0.1 % of 5# (!?!) white oil and called as “simulated condensate oil”.Overall, the “materials and methods” section needs more clarity, the formulations and components used are not clear, for example which hydrocarbon is used as oil?

Response: "Oil Contained Stability" refers to whether the stability of the foam would be affected by the presence of oil droplets present, if the oil droplets have little effect on the foam then the foam is considered to have "Oil Contained Stability". Consider of the comments raised by the reviewer, the term 'simulated oil' in the Materials section has been changed to 'experimental oil'. The oil used in the experiments was a combination of 0.9% kerosene and 0.1 % of 5# white oil.

Results and discussion

  • Point 11: Last four lines at Pag. 4. Author said: “Compared with different types of surfactants alone,” but where is reported the behavior of each foaming agents "alone"? Can they be added to figure 3?

Response: Both AOS and CHSB-35 in Figure 3 are surfactants (their chemical names were given in the "Materials" part of the revised manuscript). Based on the results in Figure 2, both surfactants are able to produce bulky foams and therefore are used in Figure 3 to compare with the new foaming agent AC-5.

  • Point 12: it would also be interesting to show the behavior of the solutions without the DP1 stabilizer, as a term of comparison!

Response: During the authors' experiments it was found that the addition of DP-1 stabiliser was effective in improving the stability of the foam (extended the half-time), but due to its low concentration it was not able to have much effect on the foam height. Considering that the focus of the manuscript was not on the composition of the AC-5 foaming agent, a blank control group was not represented in the manuscript. Also, owing to the fact that the main component of the DP-1 stabiliser was a HAPAM polymer, which could also have a significant impact on the viscoelasticity of the foam, the authors' discussion of this section focuses on the viscoelasticity explanation in the third section.

  • Point 13: Where is AC-5 system defined?

Response: To facilitate understanding, the name "AC-5" is presented in the revised manuscript under "2.2 Methods".

  • Point 14: Could you explain the phrase “It can be seen from the Figure 3 that the bubble height of the system can be increased by adding low concentrations of salt ions in a certain extent [10], while the bubble height of the system starts to decrease after further increasing the salt ion concentration.”,  I can’t understand if figure 3 and 4 are compared to show the foaming ability or somethings else.  “Bubble height” is the foaming ability or has another definition?

Response: I am sorry for the difficulty in reading this sentence, in the revised manuscript the authors have changed that sentence to "Lower concentrations of salt ions increase the volume of foam produced by foaming agents while higher concentrations have the opposite effect."

In view of the confusion created by Figures 3 and 4, the authors have decided to adopt the "foam height" as the y-axis in both figures. Thank you for your comments!

  • Point 15: “anionic HPAM” should be defined in materials and method if use or the first time you use it.

Response: This subject has been explained in the “Materials” section following the comments from reviewers.

  • Point 16: Figure 2 and more general in the text. Weigh percent is expressed as “wt.%”not “%” or other.

Response: All (including figures) incorrectly symbols in the manuscript were revised in accordance with the reviewers' comments.

  • Point 17: Figure 6. The labels (on axes and inside the graph) are too small.
  • Figure 8. The labels (on axes and inside the graph) are too small.
  • Figure 9 a and b. The labels (on axes and inside the graph) are too small, keeping into account that they will be reduced in size for publishing, it is better make them larger.
  • Also, the colors should be changed, as well as the size and symbols to improve readability.

Response: The re-edited manuscript has had the problematic images(Fig6, Fig8 and Fig 9)replaced. If you still find problems during the review process, please contact the author and thank you for your efforts!

  • Point 18: Because on the x-axis the unit is seconds, the y-axis unit should also be in seconds instead of minutes! Check “(r/min)” and “Asymptoticl” in the label inside the graph!

Response: The "mPa-s" are international units used to describe viscosity, and the "r/min" on the x-axis is one of the parameters set at the rheometer. The unit of the x-axis coordinates from "r/min" to "r/s" results in additional digits (e.g. 1 r/min - 0.0166666 r/s), on the one hand, which does not facilitate representation in the graphs , on the other hand, which reduces the accuracy of the data. Taking into account the comments of the reviewers, the authors decided to use the unit "1/60 r/s" for the x-axis.

After examining the results of the 'Asymptoticl' fit, the authors found that the results had a high correlation coefficient (0.9), which indicates that the fit results can reflect the true trend of the sample , The reason why other functions with better fitting results are not used is that foam fluids have power-law properties, thus the exponential function is more suitable than others.

  • Point 19: The draft seems interesting, in broad terms it is clear what the authors intend to describe, but unfortunately some sections are confused in the construction of the text and should be reviewed to allow the reader to understand the development of the experimental parts and the processing of the data adopted for the discussion, the "conclusion" section should also be revised.

Response: The section 'Conclusions' provides a summary of results in the manuscript, together with some expansion of potential research directions in this area. After taking into account the comments received from the reviewers, the authors decided to include the experimental procedures and data processing in 'Methods' .

  • Point 20: According to the definition of polymer as a macromolecule obtained by repeating a monomer, therefore oils and kerosene are not, which are the polymers considered/used by the authors in the work presented?

Response: The macromolecule refers to the aforementioned DP-1 stabiliser, the main component is HAPAM, which has been explained in the “Materials” section in response to comments from reviewers.

We would like to thank the referee again for taking the time to review our manuscript!

Round 2

Reviewer 2 Report

The authors have thoroughly revised the manuscript; as a result, its quality has improved compared to the previous version. In my opinion, the current version of the manuscript is acceptable for publication.

 Minor editing of English language will be good.

Reviewer 3 Report

Dear Authors,

 after your extensive revisions to the draft the article can now be published. Thank's for your job.

Best regards.